# Hybrid Chitosan–TiO_2_ Nanocomposite Impregnated in Type A-2186 Maxillofacial Silicone Subjected to Different Accelerated Aging Conditions: An Evaluation of Color Stability

**DOI:** 10.3390/nano13162379

**Published:** 2023-08-20

**Authors:** Faten K. Al-Kadi, Jwan F. Abdulkareem, Bruska A. Azhdar

**Affiliations:** 1Department of Prosthodontics, College of Dentistry, University of Sulaimani, Sulaymaniyah 46001, Iraq; jwan.abdulkareem@univsul.edu.iq; 2Nanotechnology Research Laboratory, Department of Physics, College of Science, University of Sulaimani, Sulaymaniyah 46001, Iraq; bruska.azhdar@univsul.edu.iq

**Keywords:** synthesized chitosan–TiO_2_ nanocomposite, core–shell method, room-temperature vulcanization (RTV) of maxillofacial silicone, color stability, accelerated aging conditions

## Abstract

This study explores the impact of the incorporation of a chitosan–TiO_2_ nanocomposite on the color stability of pigmented room-temperature vulcanization maxillofacial silicone under various accelerated aging conditions. Five hundred disk-shaped specimens were formed with type A-2186 silicone elastomer, and they were distributed into groups based on pigment types and nanoparticle treatments. The color difference (ΔE) was assessed using a colorimeter in the CIELAB color system before and after exposure to aging conditions, including UV-accelerated aging and outdoor weathering. ANOVA, Dennett’s T3, and Tukey HSD tests revealed significant color alterations across all silicone types, with the most pronounced being in the red-colored 3% chitosan specimens and the least pronounced being in the 2% TiO_2_ specimens that underwent UV-accelerated aging. Outdoor weathering consistently increased the ΔE values across all categories. This study suggests that while nanoparticles may offer some resistance against accelerated aging, they fall short in adequately defending against UV radiation during outdoor weathering.

## 1. Introduction

The psychosocial consequences of visible aesthetic differences among individuals are determined through a combination of individual and social factors. These differences can render individuals temporarily or permanently vulnerable, depending on the nature of the impact. The utilization of surgery or prosthetics for rehabilitation serves as a crucial means of adaptation in order to address these challenges [1].

For nearly fifty years, elastomers have been utilized in the production of facial prostheses for individuals with missing facial features. In 1986, Factor II introduced A-2186, the first commercially available platinum-catalyzed silicone elastomer. A-2186 is a pourable silicone that is clear to translucent and consists of two parts (with a base-to-catalyst ratio of 10:1) [2]. The elastomeric element of A-2186 comprises a dimethylsiloxane polymer, a reinforcing silica, and a platinum catalyst. Concurrently, the curing agent comprises a dimethylsiloxane polymer, an inhibitor, and a siloxane crosslinker. Extensive investigations conducted to compare the physical traits of A-2186 with other maxillofacial silicon variants have reported a notable increase in both the tensile strength and tear resistance of A-2186. Notably, A-2186 exhibits a viscous state at the time of loading. The heightened mechanical durability exhibited by A-2186 is likely attributable to its superior filler loading and/or the high molecular weight of the dimethylsiloxane polymer within its composition [3].

Among anaplastologists, maxillofacial prosthodontists, and dental technicians involved in fabricating facial prostheses, RTV silicone elastomer products are commonly used. A-2186 silicone, specifically, has been widely employed for intrinsic or base color packing due to its superior tensile strength, tear strength, and softer, more realistic, and skin-like surface texture [2]. 

Despite its widespread use in maxillofacial prostheses, silicone has certain limitations. One significant drawback is its short shelf life, which is primarily due to the rapid degradation of its physical properties and color stability. The overall appearance and aesthetics of a prosthesis play a crucial role in determining the patient’s wellbeing and social acceptance [4]. Clinical studies have indicated that the average lifespan of maxillofacial prostheses is approximately one and a half to two years, with many patients experiencing issues such as discoloration and decreased satisfaction within the first three years of use [5]. 

To achieve a lifelike appearance for facial prostheses, pigments are used as opacifiers and colorants for intrinsic and extrinsic coloring [2]. To create the primary skin shade, basic colors such as red, blue, and yellow are employed in maxillofacial prosthetic silicone elastomers. The amount of each color used depends on factors such as the darkness of the complexion, the complexity of the anatomy, and the desired skin tone. Red is used more for darker complexions, while blue is required more for lighter skin tones. Yellow is used to match moderate to substantial differences in skin color [6,7].

Previous studies have utilized dry earth pigments as opacifiers or intrinsic colors in their research. For instance, Beatty et al. [8] examined the color changes in dry-pigmented maxillofacial elastomer when exposed to ultraviolet light. They discovered that the red cosmetic dry earth pigment exhibited significant color changes after 400 h of exposure, while the cosmetic yellow ochre remained color-stable even after 1800 h. Similarly, Kiat-amnuay et al. [9] investigated the use of dry earth cosmetic pigments mixed with silicone A-2186 and opacifiers. Their findings aligned with those of Beatty et al. [8], demonstrating that the red pigment groups experienced the most pronounced color changes.

In recent years, significant research has been dedicated to developing a novel industrial process for incorporating nanoparticles into a polymeric matrix. This advancement has led to the emergence of a new class of polymeric materials that combine the strength of nano-oxides with the flexibility of an organic polymer matrix. Adding nanoparticles to a polymeric matrix enhances its properties due to the particles’ higher surface energy and chemical reactivity. This enables them to interact with the silicone elastomer matrix, forming a three-dimensional network within the polymer chain. By utilizing nanoparticles at the nanoscale, it becomes possible to fine-tune specific attributes of individual particles while exerting control over a range of properties, such as biological, mechanical, electrical, magnetic, and optical characteristics. For example, researchers have discovered that nanosized rutile TiO_2_ particles possess exceptional ultraviolet (UV) absorption and scattering capabilities, effectively safeguarding against UV light. These nanoparticles demonstrate diminutive dimensions, extensive surface areas, active functionality, and robust interfacial interactions with organic polymers. Consequently, they enhance the polymer’s physical and optical properties while fortifying its resilience against aging triggered by environmental stressors [10,11].

Nanoparticles serve as effective barriers against UV radiation because their size is smaller than the wavelength of UV light. When exposed to UV radiation, the electrons in nanoparticles vibrate, leading to the absorption and dissipation of a portion of the light. Consequently, the smaller the nanoparticles, the more efficient they are at shielding against solar radiation. Maintaining an appropriate filler content is crucial when incorporating nanosized oxide particles into the silicone elastomer matrix. This is due to these particles’ higher surface energy and chemical reactivity, which can lead to agglomeration if the content exceeds a certain level. The presence of agglomerated particles within the silicone elastomer matrix causes them to act as stress-concentrating centers when subjected to external forces, thus decreasing the elastomer’s mechanical strength [12].

Considering the findings of Han et al. [12] regarding the color stability of maxillofacial prostheses and previous studies on their mechanical properties, it was determined that nano-TiO_2_ at content levels of 2.0% and 2.5% proved to be the most effective opacifier when used with A-2186 silicone maxillofacial elastomer. In addition, with 2% nanosized TiO_2_, there is an increased tendency to agglomerate, as an increase in particle size and concentration results in suboptimal dispersion of nano-oxides within the matrix. An augmented propensity for agglomeration further diminishes UV-protective capabilities [10]. 

To overcome the limitations of nanoaggregation, researchers have investigated the use of TiO_2_ support material composites. By combining TiO_2_ nanoparticles (NPs) with chitosan coatings, nanocomposite films with enhanced properties, such as mechanical strength, swelling properties, and thermal stability, can be achieved. Chitosan is crucial in preventing agglomeration and facilitating the dispersion of metal oxide particles within a composite. This approach offers a potential solution for overcoming the limitations associated with nanoaggregation [13]. 

Chitosan, a natural polysaccharide derived from the chitin shells of crustaceans, is highly valued for its effectiveness in various biomedical applications. It is obtained through treating chitin with alkaline substances such as sodium hydroxide. Chitosan possesses several desirable qualities, such as biodegradability, biocompatibility, non-toxicity, and antibacterial and hydrating properties. One of its unique features is its cationic nature, which results from the presence of amino and hydroxyl groups in its structure. This cationic nature allows chitosan to interact with other compounds through electrostatic forces, through hydrogen bonds, or by incorporating the compounds into its polymeric matrix, thus enhancing its mechanical and biological properties. Furthermore, chitosan exhibits a strong affinity for negatively charged compounds, particularly under low-pH conditions. These exceptional properties, along with its biocompatibility, biodegradability, and lack of toxicity, have contributed to its widespread utilization in various fields [14,15,16]. 

Incorporating chitosan microparticles into a matrix can provide strengthening properties, as the microcracks between the matrix and chitosan can absorb energy. However, when the concentration of chitosan exceeds a certain threshold of 3.0–3.5%, the particles tend to come closer or aggregate, transforming the microcracks into macrodefects. Consequently, this results in a decrease in the material’s tensile strength [17,18]. 

Core–shell nanoparticles, which are constructed with one material encased in another, offer significant advantages in biological applications over standard nanoparticles. These benefits include reduced cytotoxicity, enhanced bio- and cytocompatibility, and improved stability, both thermally and chemically. Additionally, these nanoparticles can be more readily conjugated with other bioactive substances. The effectiveness of core–shell nanoparticles in biomedical applications is largely influenced by their surface chemistry, which enhances their ability to bond with drugs and receptors. This has inspired the creation of innovative nanoparticles that interact harmoniously with biological systems, unlike their bulk material counterparts. Furthermore, the enhanced biocompatibility and cytocompatibility of core–shell nanoparticles contribute to their therapeutic potential. This opens up exciting opportunities for the development of novel drug carriers that demonstrate improved characteristics such as longer residence times, higher bioavailability, and the ability to lower the required dosage and frequency. All of these benefits, in combination with their increased specificity, make core–shell nanoparticles a promising area of focus for future advancements in the field [19].

The hybrid composite of CS-TiO_2_ has been extensively utilized in numerous technical applications because of its potential advantages. These applications range from acting as antimicrobial agents to being used in packaging materials, and in the production of films or coatings aimed at preserving food. Furthermore, this hybrid composite has proven useful in promoting wound healing and skin regeneration, detecting glucose and alpha-fetoprotein, and even degrading water pollutants [15].

Over time, sweat and sebum can be absorbed into extraoral silicone when a prosthesis is in contact with human skin. However, the presence of UV radiation can have detrimental effects. While it increases crosslinking, it also breaks down bonds within the polymer matrix, resulting in a slower polymerization rate and silicone degradation. These processes ultimately accelerate color changes and material deterioration [20].

Silicone elastomers have been subjected to controlled experiments to replicate the effects of environmental and human conditions experienced by prostheses during their use. These experiments involve exposing the elastomers to simulated conditions, such as sebum solution or acidic perspiration, artificial daylight radiation for accelerated aging, outdoor natural weathering, and treatment with silicone cleaning solutions. By subjecting silicone elastomers to these conditions, researchers aim to closely mimic real-world scenarios and evaluate the performance and longevity of the materials used in prosthetic applications [21].

For the engineering of a color-stable prosthesis, thoroughly understanding the effects imposed by various environmental variables is of utmost importance. Although accelerated aging tests can offer indications regarding the outdoor performance of polymers and provide estimates of their service life, they can also impact the degradation mechanism and potentially yield inaccurate estimations of the actual lifespan of the polymers. Hence, it is essential to comprehensively comprehend the individual influences of environmental variables to effectively engineer a color-stable prosthesis with prolonged durability [22,23].

Although numerous research studies have focused on investigating the impact of the incorporation of different nanoparticles on the color stability of pigmented and non-pigmented maxillofacial silicone under various accelerated aging conditions [5,12,22,24,25,26,27], there is a notable absence of research exploring the effect of incorporating hybrid nanocomposites on the color stability of silicone. Specifically, not a single study has examined the influence of hybrid nanocomposites on the color stability of silicone materials.

This study aims to investigate the impact of incorporating chitosan-TiO_2_ nanocomposite on the color stability of pigmented maxillofacial silicone that has undergone room-temperature vulcanization (RTV) when exposed to various accelerated aging conditions. The null hypothesis posits that the color stability of the pigmented maxillofacial A-2186 silicone elastomer would remain unaffected by impregnation with a chitosan-TiO_2_ nanocomposite, even after undergoing different accelerated aging procedures.

## 2. Materials and Methods

### 2.1. Materials

One hundred groups (*n* = 5; 500 specimens in total) of disk-shaped specimens were prepared (twenty-two millimeters in diameter, two millimeters thick) [9,10,28,29] using A-2186 room-temperature-vulcanized (RTV) maxillofacial silicone elastomer (Factor II Inc., Lakeside, AZ, USA). The dimensions of the samples were designed using AutoCAD 2013 and processed using a CNC (Computer Numerical Control) machine (Jinan, Shandong, China). Transparent acrylic sheets were used to fabricate molds; these plastic molds were made with specified dimensions that were suitable for digital portable colorimeter devices. Each mold comprised a base, frame, and cover parts with exact measurements.

The specimens were divided into four main groups of equal size (25 specimens/group) according to the pigments that had been combined with the silicone (brilliant red, blue, and yellow dry intrinsic pigments at 0.2% concentration by weight) [12] (Technovent Ltd., Bridgend, UK), and the fourth group was without the addition of pigment (non-pigmented group). Each group was subdivided into five divisions representing the categories of silicone. Figure 1 illustrates the study design.

Five categories of silicone were made, and the silicone without incorporation of any nanoparticles was considered as the control; the second category contained titanium dioxide nanoparticles that were impregnated into the silicone at a concentration of 2% by weight [12,30] and were obtained from Sigma-Aldrich (CAS Number 718467)(Sigma-Aldrich Chemie GmbH Eschenstrasse 5 D-82024 TAUFKIRCHEN, Germany) with a primary particle size of 21 nm; the third category was impregnated with chitosan at a concentration of 3% by weight [17] and was obtained from Sigma-Aldrich (CAS Number 448869) with a 75–85% degree of acetylation and low molecular weight; the fourth category was reinforced with hybrid combination of nano-TiO_2_ and chitosan (TC) at a concentration of 1% (0.5% TiO_2_ + 0.5% chitosan) by weight; the fifth category contained silicone impregnated with a synthesized nanocomposite of chitosan-TiO_2_ powder.

Absolute ethanol 99.5% (EM- PARTA ACS; Merck-KGaA, 64271 Darmstadt, Germany) was used to assist in the dispersion of nanocomposite chitosan–TiO_2_ powder in the silicone base [31].

### 2.2. Methods

#### 2.2.1. Pilot Study

A pilot study was conducted to determine the most suitable concentration of the synthesized chitosan-TiO_2_ nanocomposite for use by evaluating its color stability after subjecting it to different media for accelerated aging. At concentrations of 0.5%, 1%, and 1.5% by weight, the chitosan–TiO_2_ nanocomposite was impregnated into red-pigmented silicone (at a concentration of 0.2% by weight of the brilliant red intrinsic pigment) and dispersed well with the assistance of absolute ethanol. The de-achromatization of weathering conditions for the red-pigmented silicone elastomers occurred at a higher rate than for those with yellow pigments and the un-pigmented samples [7,9,28,32,33,34,35,36,37].

Based on the results of the pilot study, the most suitable percentage of the chitosan-TiO_2_ nanocomposite was 1%, which resulted in the least change in the red color of the silicone after being subjecting to accelerated aging conditions.

#### 2.2.2. Preparation of the Nanocomposite

In a typical experiment with the core–shell method [38], 2 g of TiO_2_ was dispersed in 200 mL of 1% (*v/v*) acetic acid (obtained from Merck, Rahway, NJ, USA (Cat. No. 100063)) and sonicated for thirty minutes at room temperature (Q700 Sonicator; Qsonica LLC, Newtown, CT, USA), where the TiO_2_ was changed into Ti4 ions. In addition, 2 g of chitosan was dispersed in 200 mL of 1% (*v/v*) acetic acid and sonicated for thirty minutes at room temperature. Then, 200 mL TiO_2_ + 200 mL chitosan was mixed and continuously stirred until a clear solution was obtained at room temperature. NaOH solution (1 *M*) was added dropwise until the solution reached a pH of 10. The solution was then separated, and the residue was filtrated with a Buchner funnel and washed with excess distilled water thrice until a pH of 7 was obtained from the chitosan-TiO_2_ solution. After that, the mixture was dried in a vacuum oven at 60 °C overnight. Furthermore, the composite was dried using a magnetic stirrer (LabTech LMS-1003; Daihan Labtech Co., Ltd., Namyangju, Republic of Korea) for two hours at 80 °C. The dried composite was then crushed using a mortar to obtain a fine powder. An analysis of the morphological characteristics was conducted with a FESEM system, XRD, FTIR, and EDX.

#### 2.2.3. Preparation of the Control Group Specimens

According to the manufacturer’s instructions, the mixing ratio of the base to the catalyst for the RTV silicone type A-2186 was 10 to 1. The mixing of the control group began with the addition of the base to the container of an electronic balance that was sensitive to an accuracy of 0.0001 g (Nimbus Analytical, Adam Equipment, Oxford, CT, USA). The catalyst was added and mixed for five minutes with a vacuum mixer (AX-2000, Aixin Medical Equipment Co., Ltd., Tianjin, China) at a speed of 360 rpm and under a vacuum of −0.09 bar. For the intrinsically pigmented groups (brilliant red, blue, and yellow), pigment powder was added at a concentration of 0.2% [12] by weight to the container of the electronic balance, followed by the addition of the base; then, this was mixed without a vacuum for three minutes, followed by vacuum mixing for seven minutes. Next, the catalyst was added, mixed in a vacuum for the remaining five minutes, and placed in a vacuum chamber for fifteen minutes to eliminate air bubbles. Then, the mixed material was left for three minutes in the chamber without a vacuum to allow the settlement of the material. Afterward, the mixed silicone was poured into the plastic molds. G-clamps were used to tighten the mold so that the excess silicone material would flow outside its border. Then, the material was left to cure at room temperature for twenty-four hours.

#### 2.2.4. Preparation of the Experimental Group Specimens

The experimental group specimens were fabricated by combining one type of nanoparticle (TiO_2_, chitosan, and the hybrid chitosan–TiO_2_ hybrid nanocombination (TC)) at 2%, 3%, and 1%, respectively, either alone or with one of the three dry intrinsic pigments (brilliant red, blue, and yellow) at a concentration of 0.2% [12] by weight; the base of the silicone to be mixed was as same as that in the control specimens, as stated in the manufacturer’s instructions.

#### 2.2.5. Preparation of the Experimental Group Specimens Reinforced with the Synthesized Chitosan–TiO_2_ Nanocomposite

The synthesized chitosan–TiO_2_ nanocomposite powder was combined with ethanol at weight percentages to fabricate the specimens. The volume of ethanol used depended on the amount of silicone base required to create the five specimens. The ethanol and chitosan–TiO_2_ nanocomposite mixture was subjected to sonication using a Q700 Sonicator from Qsonica LLC. The sonication process lasted for thirty minutes at room temperature with continuous cooling. A titanium alloy probe with a standard half-diameter of 136 × 13 mm was utilized, and an amplitude power of four hundred watts was employed with a pulse-on time of five seconds and a pulse-off time of two seconds.

Next, the synthesized chitosan-TiO_2_ nanocomposite and ethanol mixture was added to the RTV silicone type A-2186 base to prepare the specimens of the experimental group. The mixture was thoroughly mixed in a vacuum mixer for ten minutes. The vacuum mixer jar was placed on a magnetic hotplate stirrer (LabTech LMS-1003; Daihan Labtech Co., Ltd.) and connected to a vacuum rotary pump (EuroVac; Thompson CSF, La Défense, France) to evaporate the ethanol under a pressure of −0.075 MPa. The evaporation process took 120 min. To ensure the homogeneous dispersion of the synthesized chitosan–TiO_2_ nanocomposite within the silicone, the concoction was mixed every three minutes for one minute during the two-hour period. Subsequently, the mixture was cooled to room temperature before the catalyst was added. The mixture was mixed for an additional five minutes in the vacuum mixer. Finally, the prepared mixture was loaded into molds and transferred to a vacuum chamber for two minutes to eliminate any air bubbles. G-clamps were used to tighten the mold so that the excess silicone material would flow outside its border. Then, the material was left to cure at room temperature for twenty-four hours. All of the specimens were then trimmed and cut-marked to classify the groups.

#### 2.2.6. Color Stability Test

The prepared specimens were subjected to a color stability test that was carried out using a colorimeter (Portable Color Analyzer, Precise Digital Colorimeter/Color Meter Lab Tester with a four-millimeter measurement aperture—FRU WR10QC, China).

The evaluation of color differences was performed using the CIELAB (Commission Internationale de l’Eclairage) colorimetric system, which is shown in Figure 2. This system utilizes three parameters to define color: L*, a*, and b*. The L* axis corresponds to brightness, ranging from zero (black) to one hundred (pure white). The a* coordinate indicates the presence of red (positive values) or green (negative values), while the b* coordinate represents the amount of yellow (positive values) or blue (negative values).

The CIE colorimetric system allows for the calculation of the average ∆E (color variation) between two measurements using a specific formula:ΔΕ = [(ΔL*) + (Δa*) + (Δb*)]^1/2^(1)

This formula enables the determination of the mean color variation (∆E) value by comparing the L*, a*, and b* values of two readings. Equation (1) was used.

Before weathering, all test specimens were washed, cleaned with distilled water, and gently dried with a clean paper towel. When readings were taken, a white card served as the background [39]. The L*, a*, and b* color coordinates for each sample prior to exposure to weathering were documented. Three measurements were taken and averaged, and the mean for each sample was determined using the CIELAB uniform color scale. L*, a*, and b* were the differences in the respective values before and after aging.

In the context of the LAB color space, the ΔE measure, which is determined as the Euclidean distance between two LAB vectors, is commonly employed to assess color perception. A lower value suggests that it is harder for an observer to differentiate between two colors, with 3.00 as the threshold value for human color discrimination.

Values of ∆E between zero and one represent unnoticeable color differences, whereas values between two and three represent color differences that are just noticeable. When values of ∆E exceed or are equal to 3.3, the color difference is visually noticeable and clinically unacceptable.

Distinguishing between perceptible and acceptable color changes is common in various studies. The variation in acceptability thresholds among these studies can be attributed to the variations in pigments and silicones used. In the context of this study, alterations in coloration that are below three units of magnitude are perceived to be visually detectable and are regarded as clinically tolerable [40].

#### 2.2.7. Conditioning Modes

The specimens were randomly assigned to five groups, with each representing a different conditioning method. The color stability of each of the five categories of pigmented and non-pigmented silicone was evaluated according to the color difference before and after they were subjected to five different accelerated aging conditions. The first of these conditions was storage in a commercial antibacterial silicone-cleaning solution (B-200-12, Daro Inc., Lakeside, AZ, USA) for thirty hours. The second condition was storage in a simulated sebum solution for six months; the sebum was prepared by dissolving 10% palmitic acid with 2% glyceryl tripalmitate in 88% linoleic acid (all *w/w*) [30,40,41,42]. The third condition was storage in simulated acidic perspiration (sweat) for six months; the sweat was prepared according to the specifications of the International Organization for Standardization (ISO 105-E04:96) [43] and comprised the following components (per liter of distilled water): 0.5 g of L-histidine monohydrochloride monohydrate, 5 g of sodium chloride, and 2.2 g of sodium dihydrogen orthophosphate dehydrate [30,40,41,42]. The fourth group of specimens was subjected to exposure to accelerated weathering with artificial ultraviolet (UV) radiation for 720 h. This was facilitated by an accelerated UV aging chamber equipped with two lightbulb fixtures positioned directly above the specimens. These fixtures provided ultraviolet light exposure equivalent to 720 KJ/m^2^/h. In addition, the aging chamber was maintained at a consistent temperature of 60 °C and relative humidity of 80% [22,32]. The fifth group of specimens was subjected to outdoor weather conditioning for six months.

The environmental exposure of the samples to atmospheric conditions was conducted in accordance with the standards set forth in Designation G7-8.3.1 [44] of the American Society for Testing and Materials. The specimens that underwent natural outdoor weathering were meticulously suspended from stainless steel racks with ligature wire. The entire assembly was deliberately positioned on the roof of the College of Dentistry, University of Sulaimani, for six months [30,40,45,46] from February 2022 to August 2022. The monthly average high and low temperatures and climatic data were documented during the outdoor weathering, as shown in Table 1. The specimens were left uncovered and exposed to environmental conditions during the weathering exposure. They were suspended in such a way that there were no barriers or walls from the back, front, right, or left side, thus ensuring optimal exposure to environmental conditions. The specimens were checked daily to ensure that their position was not changed. Before the specimens were evaluated, they were cleaned for ten minutes in distilled water, wiped dry, and tested.

The durations of conditioning were chosen to mimic the use of silicone prostheses over a period of twelve to eighteen months. Typically, patients use their prosthesis for eight to twelve hours a day; during this time, the prosthesis is anticipated to be exposed to one to three hours of daylight, regular environmental conditions, and constant sebum and sweat when worn on the defect site. Furthermore, patients typically dedicate around five minutes to cleansing their prostheses before bedtime. Hence, the duration of one month of service corresponds to approximately thirty to ninety hours [22] of daylight aging, along with ten to fifteen days of storage in sebum or acidic solutions (sweat), and one hundred and fifty minutes of storage in cleaning solutions [40].

### 2.3. Statistical Analysis

A comprehensive statistical analysis was carried out to concisely summarize the findings related to each variable that was studied. Descriptive statistics were calculated and presented for continuous variables, including the mean and standard deviation. A one-way analysis of variance (ANOVA) was utilized with a significance level of *p* < 0.05 to evaluate significant differences among the five accelerated aging conditions across the five categories of silicone specimens.

Since ANOVA operates on the assumption of equal variances across specimens, Levene’s test of homogeneity of variance (α = 0.05) was applied to all data, assuming equal variance. Consequently, when the equal-variance assumption was rejected, Dennett’s T3 multiple comparison test was used to compare the groups (*p* < 0.05). On the other hand, when the assumption of equal variance was accepted (*p* < 0.05), Tukey’s HSD multiple comparison tests were used to compare the groups.

A one-sample *t*-test was performed within each accelerated aging condition to examine the effects of the modes on color change, with a significance level of *p* < 0.05.

The finding that the variable in this analysis was normally distributed was obtained by applying the Shapiro–Wilk and Kolmogorov–Smirnov tests. Then, a *t*-test and one-way ANOVA were performed, and version 27.0 of the SPSS program for Windows (SPSS for Windows) was utilized.

## 3. Results

### 3.1. Morphological Characteristics

#### 3.1.1. Scanning Electron Microscopy (SEM)

SEM was used to study the surface morphologies of the various coatings of nanoparticles; the SEM analysis revealed that the average particle size of the synthesized chitosan-TiO_2_ nanocomposite was from 31.52 to 60.07 nm. Figure 3. Field-emission scanning electron microscopy (FESEM) was used to acquire information about the external morphology of the nanocomposites. In this research, FESEM confirmed the adequate homogenous distribution of nanosized and less-aggregated TiO_2_ particles within the chitosan (CS) matrix. The irregular spherical morphologies were due to the interlocking interaction between TiO_2_ and CS, which consequently improved the homogeneity of the two.

#### 3.1.2. X-ray Diffraction (XRD)

The crystalline structure of the synthesized nanocomposites was analyzed via XRD. XRD patterns of the synthesized chitosan-TiO_2_ nanocomposite, pure TiO_2_, and chitosan are demonstrated in Figure 4. The XRD pattern of chitosan displayed small, broad peaks at (11.8°) and (20.0°), indicating its semi-crystalline structure. This suggests that the incorporation of TiO_2_ into the chitosan matrix predominantly takes place in the semi-crystalline region of chitosan. In contrast, both chitosan and TiO_2_ exhibited distinct diffraction peaks in the XRD pattern of the nanocomposite. The TiO_2_ nanoparticles displayed the presence of anatase and rutile forms, as evidenced in Figure 4c. The coexistence of these mixed phases of TiO_2_ is advantageous in minimizing the recombination of photogenerated electrons and holes, thereby enhancing the photocatalytic activity of titanium.

#### 3.1.3. Fourier-Transform Infrared Spectroscopy (FTIR)

The FTIR analysis, Figure 5, revealed the presence of characteristic bands corresponding to chitosan, TiO_2_, and the chitosan-TiO_2_ nanocomposite. In the FTIR spectra of chitosan, Figure 5a, the peaks around (915, 1029), (1250), (1340, 2888), (1550), and (3460) cm^−1^ are related to (C-O-C), (C-O-H), (C-H), (C=O), and (OH−), respectively. In the FTIR spectra of TiO_2_, Figure 5b, the peaks around (530, 654) are assigned to the bending vibration of (Ti-O-Ti) bonds, while that at 3500 cm^−1^ corresponds to the hydroxyl group (OH−).

Figure 5c,d demonstrate the FTIR spectra of the synthesized nanocomposite of chitosan-TiO_2_. In the spectra, the peaks around (3441, 3437), (2855, 2926), and (1621, 1619) cm^−1^ are related to (O–H), (C–H), and (C=O) groups, respectively. The peaks around (1575, 1500), (1425, 1352), and (1112, 1040) cm^−1^ are related to (N–H), (CH–OH), and (CH2–OH) groups, respectively. The fingerprint band between 700 and 400 cm^−1^ shows stretching vibrations of (Ti-O-Ti), which indicates the immobilization of TiO_2_ onto the chitosan matrix.

#### 3.1.4. Energy-Dispersive X-ray (EDX)

Energy-dispersive X-ray (EDX) spectroscopy is frequently utilized to determine the elemental distribution within a material’s composition. In this research, the presence of TiO_2_ within the nanocomposite was examined using EDX. The existence and dispersion of TiO_2_ throughout the composite were confirmed via the EDX spectrum, as shown in Figure 6. From the data presented in Table 2, it can be inferred that TiO_2_ has been incorporated into the composite, and its particulates have been uniformly dispersed.

### 3.2. Pilot Study

The mean, standard deviation, and *p*-value for ΔE in the pilot study are listed in Table 3; the red-pigmented experimental silicone specimens that were modified with 1% chitosan-TiO_2_ and chitosan-TiO_2_ 1.5% nanocomposites exhibited significantly less of a color change (*p* < 0.05) than the chitosan-TiO_2_ 0.5% specimens did when exposed to outdoor weather for six months.

Moreover, as shown in the statistical analysis, the ΔE of the red color in the 1% chitosan–TiO_2_ specimens was found to be significantly lower (*p* < 0.05) than that in the 0.5% chitosan–TiO_2_ and 1.5% chitosan–TiO_2_ specimens when subjected to sebum. These findings led to the selection of the 1% chitosan–TiO_2_ nanocomposite as the percentage for use in the specimens of the subsequent experimental group.

### 3.3. Color Stability Results

The mean, standard deviation, and significant differences (*p*-value) for the brilliant red, blue, yellow, and non-pigmented ΔE among all silicone categories after being subjected to different conditions are demonstrated in Table 4, Table 5, Table 6 and Table 7, respectively.

#### 3.3.1. The Red Color

For the brilliant red pigments, within all silicone specimens, the UV weather (720 h) induced a prominent alteration in color (ΔE = 27.93, 31.38, 27.36, 41.19, and 40.84) (*p* < 0.05), with the greatest ΔE being observed in the 3% chitosan silicone specimens (ΔE = 41.19). However, smaller color changes were observed in all silicone specimens after immersion in antibacterial cleaning solution (30 h), with the smallest color changes being observed for 2% TiO_2_ and 1% TC (ΔE = 1.03 and 1.35), respectively.

According to the statistical analysis, all conditioning modes provoked visually detectable color changes (ΔE > 3) in the 1% chitosan–TiO_2_, 3% chitosan, and control (zero nanocomposite) silicone specimens. The color changes in all specimens were significantly smaller (*p* < 0.05) than those in the control (zero nanocomposite) specimens when they were exposed to sebum. Consistently, the ΔE values of the 1% TC and 2% TiO_2_ categories were significantly lower (*p* < 0.05) than those of the 3% chitosan and control (zero nanocomposite) silicone specimens when exposed to the antibacterial cleaning solution.

**Table 4 nanomaterials-13-02379-t004:** Mean values and standard deviations of the ΔE values of the brilliant red silicone categories under different conditions.

Groups	Brilliant Red ΔE					
	1% Chitosan–TiO_2_	1% TC	2% TiO_2_	3% Chitosan	Control (Zero Nanocomposite)	*p*-Value
Sweat (6 months)	6.66 ± 3.60	6.14 ± 0.38	5.45 ± 0.58	5.01 ± 2.02	5.15 ± 1.43	0.561
Antibacterial cleaning solution (30 h)	3.31 ± 1.67	1.35 ± 0.49 ^c,d^	1.03 ± 0.98 ^c,d^	4.62 ± 2.39	4.66 ± 1.11	0.000
Outdoor weather (6 months)	16.61 ± 6.32	21.19 ±1.70	19.74 ± 2.95	20.87 ± 1.03	18.91 ± 9.90	0.695
UV weather (1 month 720 h)	27.93 ± 1.24 ^a,c^	31.38 ± 0.6 ^b,c^	27.36 ± 0.71 ^c^	41.19 ± 2.6	40.84 ± 7.15	0.000
Sebum (6 months)	6.24 ± 1.66 ^d^	7.48 ± 1.66 ^d^	6.25 ± 2.34 ^d^	9.26 ± 1.85 ^d^	14.44 ± 2.36	0.000
*p*-value	0.000 *	0.000 *	0.000 *	0.000 *	0.000 *	

^a^: In comparison with 1% TC. ^b^: In comparison with 2% TiO_2_. ^c^: In comparison with 3% chitosan. ^d^: In comparison with the control (zero nanocomposite). Different superscript letters in the same row indicate that significant differences in ΔE (*p* < 0.05) were present only after applying one-way ANOVA, Dennett’s T3 test, and Tukey’s HSD multiple comparison tests. * There were overall significant differences among conditions in the color changes in all silicone specimens.

#### 3.3.2. The Blue Color

Significant color changes (ΔE = 9.08, 8.92, 10.65, 7.73, and 5.71) (*p* < 0.05) were most remarkable in the blue-pigmented specimens across all silicone categories when subjected to outdoor weather for six months. Specifically, the 2% TiO_2_ specimens exhibited the greatest color changes (ΔE = 10.65). However, when these 2% TiO_2_ specimens were exposed to sweat for six months and an antibacterial cleaning solution for thirty hours, they displayed significantly smaller color changes (ΔE = 0.96 and 0.74, respectively).

However, when exposed to all tested conditions except for the antibacterial cleaning solution, the 1% chitosan–TiO_2_ and 3% chitosan specimens showed visually detectable color changes (ΔE > 3).

Statistically, the color change, as indicated by the ΔE value, of the 1% chitosan–TiO_2_ specimen was significantly greater (*p* < 0.05) than that in the control (zero nanocomposite) specimens when exposed to UV weathering for 720 h. Similarly, the color change in the 3% chitosan specimens was significantly greater (*p* < 0.05) than that in the control (zero nanocomposite) specimens when exposed to sweat, outdoor weathering, and UV weathering for 720 h.

On the other hand, 1% TC and 2% TiO_2_ specimens displayed significantly smaller color changes (*p* < 0.05) than the 3% chitosan specimens when exposed to both sweat and sebum.

**Table 5 nanomaterials-13-02379-t005:** Mean values and standard deviations of ΔE for the blue silicone categories under different conditions.

Groups	Blue ΔE					
	1% Chitosan-TiO_2_	1% TC ^a^	2% TiO_2_ ^b^	3% Chitosan	Control (Zero Nanocomposite)	*p*-Value
Sweat (6 months)	3.49 ± 2.57	1.18 ± 0.82 ^c^	0.96 ± 0.26 ^c^	3.46 ± 1.02 ^d^	1.50 ± 0.55	0.001
Antibacterial cleaning solution (30 h)	1.21 ± 0.67	1.37 ± 0.93	0.74 ± 0.43 ^c^	2.94 ± 1.39	1.32 ± 0.31	0.001
Outdoor weather (6 months)	9.08 ± 1.44 ^d^	8.92 ± 0.59 ^d^	10.65 ± 0.25 ^d,c^	7.73 ± 0.66 ^d^	5.71 ± 1.57	0.000
UV weather 1 month (720 h)	3.73 ± 0.89 ^d^	2.67 ± 0.58	2.67 ± 0.37	3.63 ± 0.93 ^d^	1.88 ± 0.86	0.005
Sebum (6 months)	4.21 ± 1.79	2.84 ± 1.34 ^c^	3.09 ± 1.36 ^c^	5.76 ± 2.19	4.85 ± 1.26	0.018
** *p* ** **-value**	0.000 *	0.000 *	0.000 *	0.000 *	0.000 *	

^a^: In comparison with 1% TC. ^b^: In comparison with 2% TiO_2_. ^c^: In comparison with 3% chitosan. ^d^: In comparison with the control (zero nanocomposite). Different superscript letters in the same row indicate that significant differences in ΔE (*p* < 0.05) were present only after applying one-way ANOVA, Dennett’s T3 test, and Tukey’s HSD multiple comparison tests. * There were overall significant differences among conditions in the color changes in all silicone specimens.

#### 3.3.3. The Yellow Color

Among all of the silicone specimens with yellow pigment, the most significant color change was observed when they were subjected to outdoor weather for six months. The highest ΔE value of 3.04 was recorded for the control (zero nanocomposite) specimens. Conversely, the 2% TiO_2_ category specimens displayed the smallest color change after being subjected to sweat, with a ΔE value of 0.48.

In the statistical analysis, it was found that under most of the conditions, excluding that of sweat, there was not a significant color change in any of the specimens when they were compared with each other. However, when exposed to sweat, a significantly greater color change (*p* < 0.05) was seen in the 1% chitosan–TiO_2_ and 1% TC specimens than in the 2% TiO_2_ specimen. In contrast, with the same sweat exposure, the 2% TiO_2_ specimen showed a significantly lower ΔE value (*p* < 0.05) than both the 3% chitosan and control (zero nanocomposite) specimens; the observed color changes were within the clinically accepted limits, with ΔE < 3.

**Table 6 nanomaterials-13-02379-t006:** Mean values and standard deviations of the ΔE values of the yellow silicone categories under different conditions.

Groups	Yellow ΔE					
	1% Chitosan–TiO_2_	1% TC ^a^	2% TiO_2_	3% Chitosan	Control (Zero Nanocomposite)	*p*-Value
Sweat (6 months)	1.34 ± 0.48 ^b^	1.69 ± 0.85 ^b^	0.48 ± 0.23 ^c,d^	1.5 ± 0.60	2.21 ± 1.24	0.006
Antibacterial cleaning solution (30 h)	1.84 ± 0.50	1.14 ± 0.35	0.86 ± 0.38	1.41 ± 0.74	1.45 ± 0.78	0.103
Outdoor weather (6 months)	2.61 ± 0.56	2.64 ± 0.63	2.72 ± 0.74	2.72 ± 0.40	3.04 ± 0.61	0.803
UV weather (1 month; 720 h)	0.95 ± 0.26	0.7 ± 0.26	1.02 ± 0.16	0.89 ± 0.32	1.13 ± 0.43	0.267
Sebum (6 months)	1.07 ± 0.36	1.49 ± 0.87	1.38 ± 0.50	1.32 ± 0.40	1.6 ± 1.29	0.788
*p*-value	0.000 *	0.002 *	0.000 *	0.000 *	0.035 *	

^a^: In comparison with 1% TC. ^b^: In comparison with 2% TiO_2_. ^c^: In comparison with 3% chitosan. ^d^: In comparison with the control (zero nanocomposite). Different superscript letters in the same row indicate that significant differences in ΔE (*p* < 0.05) were only present after applying one-way ANOVA, Dennett’s T3 test, and Tukey’s HSD multiple comparison tests. * There were overall significant differences among conditions in the color changes in all silicone specimens.

#### 3.3.4. The Non-Pigmented Group

In the non-pigmented group, the most substantial color changes across all silicone categories occurred when the specimens were exposed to sebum, with the control (zero nanocomposite) specimens showing a ΔE value of 9.02. On the other hand, the UV weathering for 720 h resulted in significant color changes across all categories when they were compared with each other. Interestingly, the antibacterial cleansing solution induced the smallest color change across all silicone categories, with a low ΔE value of 0.76 being recorded for the 2% TiO_2_ silicone variant.

From the statistical analysis, it emerged that the silicone specimens in the 3% chitosan category had a significantly greater color change (*p* < 0.05) than those in the control (zero nanocomposite) category when exposed to sweat for six months and UV weathering for 720 h. However, when exposed to outdoor weathering for six months, the specimens in the 1% chitosan–TiO_2_ category exhibited a significantly higher ΔE value (7.77) (*p* < 0.05) than those of all other silicone categories.

Moreover, when exposed to sebum, the color change in the 1% chitosan–TiO_2_ and 1% TC categories was significantly higher (*p* < 0.05) than that in the 2% TiO_2_ silicone category. However, when exposed to the same conditions, the 2% TiO_2_ category was found to have a lower ΔE value (*p* < 0.05) than the control (zero nanocomposite) silicone category. Additionally, with exposure to sweat, the ΔE values of all specimens were significantly lower (*p* < 0.05) than those of the 3% chitosan silicone category.

**Table 7 nanomaterials-13-02379-t007:** Mean values and standard deviations of the ΔE values of the non-pigmented silicone categories under different conditions.

Groups	Non-Pigmented ΔE					
	1% Chitosan–TiO_2_	1% TC	2% TiO_2_	3% Chitosan	Control (Zero Nanocomposite)	*p*-Value
Sweat (6 months)	1.12 ± 0.41 ^c^	1.70 ± 0.99 ^c^	1.26 ± 0.59 ^c^	7.63 ± 3.5 ^d^	1.39 ± 0.46	0.000
Antibacterial cleaning solution (30 h)	1.24 ± 0.48	1.10 ± 0.37	0.76 ± 0.37	1.17 ± 0.32	0.95 ± 0.39	0.294
Outdoor weather (6 months)	7.77 ± 0.36 ^a,b,c,d^	3.42 ± 0.45	4.11 ± 0.93	3.13 ± 1.61	4.67 ± 2.54	0.000
UV weather (1 month 720 h)	3.81 ± 0.35 ^a,b,d^	2.00 ± 0.51 ^c^	1.8 ± 0.34 ^c^	5.04 ± 1.21 ^d^	1.88 ± 0.99	0.000
Sebum (6 months)	4.81 ± 0.92 ^b^	4.84 ± 0.92 ^b^	2.55 ± 0.52 ^d^	5.5 ± 4.28	9.02 ± 3.96	0.009
*p*-value	0.000 *	0.000 *	0.000 *	0.014 *	0.000 *	

^a^: In comparison with 1% TC. ^b^: In comparison with 2% TiO_2_. ^c^: In comparison with 3% chitosan. ^d^: In comparison with the control (zero nanocomposite). Different superscript letters in the same row indicate that significant differences in ΔE (*p* < 0.05) were only present after applying one-way ANOVA, Dunnett’s T3 test, and Tukey’s HSD multiple comparison tests. * There were overall significant differences among conditions in the color changes in all silicone specimens.

## 4. Discussion

In this study, all silicone categories, whether pigmented or non-pigmented, underwent different amounts of color change regardless of their aging conditions. The results of this study support the rejection of the null hypothesis that the color stability of the pigmented maxillofacial A-2186 silicone elastomer was affected by impregnation with the chitosan–TiO_2_ nanocomposite after being subjected to different accelerated aging conditions.

The deterioration of color in facial prostheses can be attributed to environmental factors, such as solar radiation, temperature, and water. Solar radiation consists of ultraviolet (UV), visible, and infrared radiation, with UV radiation significantly impacting color stability. The depletion of the ozone layer since the 1970s has raised concerns about the effects of UV radiation on facial prostheses [12].

Additionally, routine cleaning and disinfection procedures can lead to color alterations in maxillofacial silicone prostheses due to the high permeability of silicone. While various cleansing agents, such as water, neutral soap, and chlorhexidine, are recommended, they should be used cautiously to prevent adverse effects on the material’s physical properties [47].

In alignment with the study by Hatamleh et al. [40], the findings of this research also demonstrated that the antibacterial cleaning solution ensured the greatest color stability across all silicone categories, regardless of whether they were pigmented or not.

This study introduced the development of a novel three-phase composite by merging nanoparticles with polymeric silicone, a technique that has not been previously employed. This composite was successfully created by meticulously integrating two different particles—nanoparticles (TiO_2_) and microparticles (chitosan)—in specific proportions. As a result, the overall qualities of the silicone polymers saw significant enhancements. This progressive development is a substantial stride toward endowing maxillofacial prostheses with ideal properties and realistic attributes.

Hybrid nanoparticles, such as chitosan–TiO_2_ composites, have generated significant interest, as they merge the properties of organic and inorganic components, resulting in novel materials with enhanced and unique characteristics [15,48].

The core–shell mixing method was effectively employed in this study to prepare a TiO_2_-supported chitosan nanocomposite. Comprehensive morphological characterization utilizing scanning electron microscopy (SEM) (Figure 3), X-ray diffraction (XRD) (Figure 4), and Fourier-transform infrared spectroscopy (FTIR) (Figure 5) confirmed the successful adsorption of the TiO_2_ nanopowder into the chitosan matrix with excellent dispersion. These findings demonstrate the meticulous preparation and integration of the TiO_2_ and chitosan components within the nanocomposite.

Achieving proper dispersion of chitosan–TiO_2_ nanopowder in a polymer matrix, such as silicone, poses a significant challenge in nanocomposite production due to nanoparticle aggregation. However, achieving uniform dispersion is crucial for polymers and nanocomposites that require enhanced color stability. Ethanol, as a polar solvent that possesses hydroxyl (OH) groups, exhibits strong reactivity with ions, resulting in a prolonged dispersion effect. Its use as a stabilizer facilitates the sustained dispersion of a nanopowder within the ethanol solvent, thereby enabling a more uniform dispersion within the polymer matrix [48].

A novel method was employed in this study to achieve improved dispersion of synthesized chitosan–TiO_2_ nanocomposite powder within a silicone matrix. The nanocomposite powder was dispersed in ethanol through sonication and subsequently mixed with the silicone base. The silicone mixture was subjected to heating and vacuuming to remove the ethanol. This approach intended to achieve a higher level of dispersion without the need for any additional third-party materials that could potentially compromise the properties of the silicone. According to Abdalqadir et al. [31], ethanol was found to have no detrimental effects on the structure and integrity of silicone elastomers.

Based on the current study’s findings, the red-colored samples in all silicone categories showed drastic color alterations after undergoing 720 h of UV-accelerated aging. The most substantial change was observed in the 3% chitosan samples (41.19), and the smallest was observed in the 2% TiO_2_ samples (27.36). Similarly, outdoor weathering over a period of six months led to a steep increase in ΔE values, which exceeded 21.19.

From a visual perspective, the pigments transitioned from a brilliant red to a lighter pink, and the pigmented elastomers indicated near-total color depletion. The findings are analogous to those of prior research conducted by Beatty et al. [8] and Kiat-Amnuay et al. [9].

Subsequently, the ΔE values for the brilliant-red pigment in all silicone categories exhibited discernible color shifts (ΔE > 3) in each instance of the accelerated aging processes, apart from the ΔE values of 2% TiO_2_ and 1% TC, which demonstrated minimal color transformations after being submerged in an antibacterial solution for thirty hours, registering values of 1.03 and 1.35, respectively.

These observations align with the findings of earlier research by Beatty et al. [8] and Kiat-Amnuay et al. [33], who pointed out that using an intrinsic red pigment may contribute to color degradation. This could be due to the organic nature of such red pigments, which are more susceptible to the effects of irradiation. Organic colorants, depending on whether double and triple bonds are used for color provision, tend to be relatively reactive and less stable. Conversely, inorganic pigments usually offer greater color stability. However, they are often not favored due to their lower brightness compared with that of organic pigments, thus presenting a challenge in achieving a good color match.

Silicone is a polymer that is commonly available in the form of a moderately viscous liquid and is characterized by weak molecular interactions and difficulties in incorporating pigments. This property leads to chromatic alterations in the material, as smaller pigment particles tend to aggregate, while larger particles tend to separate from the polymer. Moreover, the polymerization process of silicone elastomers can be influenced by organic materials, such as makeup powder. Consequently, the presence of pigments may delay the polymerization process and exacerbate the effects of accelerated aging on the materials [11].

Among the various pigments, the cosmetic powder displayed the greatest degree of color alterations. This phenomenon could be attributed to factors such as the incorporated particles’ magnitude or the impact of aging. Silicone, which is known for its lower cohesive energy, exhibits weaker molecular interactions. Consequently, diminutive particles have a propensity to cluster, while their larger counterparts often disengage from the polymer, providing no reinforcement for the material’s structure. The particulate components of the cosmetic powder, which were possibly of a larger size, were more prone to separation from the polymer chain, which could have contributed to the increased color instability within these materials. Additionally, pigments derived from organic sources, such as makeup powder, undergo amplified degradation with aging, often dissolving upon interaction with ultraviolet light [27]. In addition, due to their larger size, dry earth pigments, which bear a structural similarity to cosmetic powders, tend to remain dispersed instead of becoming integrated within the polymer matrix. This characteristic potentially increases their vulnerability to UV degradation [44].

The susceptibility of polymeric biomaterials to deterioration under environmental conditions stems from their inadequate resistance to significant thermal changes and sunlight. Elements of weather, such as temperature, solar radiation, and moisture, can impact the properties of silicone elastomers by triggering chemical alterations. These changes, in turn, result in modifications to the functional features of these materials. The effective performance of silicone elastomers in response to extraoral factors can be ascertained by conducting tests that accurately simulate the conditions associated with outdoor exposure [45].

This study revealed that both outdoor weathering and UV-accelerated artificial weathering had a notable impact on the color of all types of silicone, irrespective of whether they were pigmented or not. Silicone, as a polymer, possesses aromatic rings and C=C bonds, which can be susceptible to the effects of UV light, resulting in color instability. When these functional groups in a polymer absorb UV light, they become energetically unstable. However, this excess energy can be mitigated through various means, such as by transferring the excitation to other molecules for stabilization. The excited groups can then return to their original state by releasing the excess energy in the form of light with a longer wavelength or heat. Failure to dissipate the excess energy can lead to photochemical degradation, causing detrimental effects such as color or brightness loss, reduced opacity, and material stiffness. Consequently, the presence of aromatic rings and C=C bonds in polymers allows UV-light-induced degradation, leading to adverse changes such as altered color and brightness, decreased opacity, crack formation, and increased rigidity [22,49].

As demonstrated by the findings of this study, the samples that underwent artificial aging showed more considerable color changes (ΔE) in comparison with those that were exposed to outdoor weathering. In outdoor weathering, the samples were subjected to natural conditions, yet this approach lacked the ability for precise control and was deemed subjective. In such weathering experiments, accurate regulation cannot be achieved for factors contributing to degradation, including geographic location, seasonal changes, specific weather conditions, time of day, and exposure duration [22].

The observed outcome could be attributed to the occurrence of post-polymerization crosslinking triggered by light irradiation, which leads to alterations in the structure of the polymer network. These modifications may involve changes in the polymer chain length, intermolecular bonding, and spatial arrangement of the polymer chains. Consequently, these changes affect light transmission through the maxillofacial material and contribute to the degradation of the polymer’s color shade [22,49].

This study declares that integrating nanosized particles, whether they were utilized independently or as a composite, exhibited insufficient efficacy in safeguarding silicone against color degradation. The influences of all types of weathering on silicone color, particularly under outdoor weathering conditions, were notably significant.

Nano-TiO_2_ is commonly used as an inorganic UV absorber due to its high thermal stability and photostability, unlike organic UV absorbers, which tend to migrate within the polymeric matrix and are less stable. When nanoparticles are exposed to UV light, their electrons vibrate, leading to a combination of scattering and absorption of UV radiation. The UV-shielding ability of nanoparticles is a result of this combined effect. Smaller particle sizes and lower nano-oxide concentrations enhance their dispersion within the elastomer matrix, thereby improving UV shielding [10,12,50].

According to Bangera et al. [51], aggregates of nano-oxide particles typically range between 30 and 150 nm, representing the most minor units in sunscreen formulations. These aggregates are formed through clustering primary particles, creating tightly bound structures that are larger than the individual building blocks. Increased particle size and concentration result in poor dispersion and a higher tendency for particle agglomeration, reducing the effectiveness of UV shielding. This is supported by studies conducted by Akash et al., Han et al., and Bishal et al. [10,12,50].

The decline in color stability observed in this study following exposure of the elastomer to outdoor conditions can be attributed to the incorporation of a nanosized composite comprising chitosan–TiO_2_ particles with sizes ranging from 31 to 60 nm. The aggregation of these particles played a significant role in this reduced stability. These findings align with the research conducted by Bangera et al. [51], further supporting the notion that particle aggregation negatively impacts color stability in silicone elastomers subjected to outdoor conditions.

All silicone categories, whether pigmented or non-pigmented, demonstrated considerable color alterations upon exposure to outdoor weathering. Notably, the greatest color modification was detected in the blue-pigmented silicone with a ΔE value of 10.65—specifically, in the 2% TiO_2_ variant. Similarly, significant color shifts were observed in the yellow-pigmented silicone, with the control variant registering a maximum ΔE value of 3.04. Furthermore, non-pigmented silicone was not immune to these changes; the chitosan–TiO_2_ variant showed the most substantial color change, with a ΔE value of 7.77. These findings underline the broad impact of outdoor weathering on color stability across all silicone types.

The exposure of elastomers to outdoor conditions led to a slight whitening and yellowing phenomenon. This was caused by the photo-oxidative degradation of the polymer, which occurred due to the combined action of oxygen and sunlight. The process involves the formation of free radicals, reactions with oxygen, and the subsequent production of polymer oxy and peroxy radicals, resulting in chain scission. Additionally, crosslinking can occur through reactions between different free radicals or bonding between existing monomers and chains. When a polymer molecule absorbs ultraviolet light, it becomes molecularly unstable. Excess energy can be transferred between molecules, allowing the molecules to regain stability. However, the release of excess energy leads to photochemical degradation, contributing to the deterioration of the polymer molecules. This degradation process causes modifications in the polymer network structure, thus affecting the number of polymer chain units, their bonding, and their spatial arrangement. Consequently, this affects light transmission through the maxillofacial material and degrades the polymer’s color shade. These findings are supported by research conducted by Malavazi et al. [52] and Hatamelh et al. [40], who emphasized the significance of photochemical degradation and crosslinking in the color changes observed in polymers exposed to outdoor environments.

In addition, facial prostheses are in direct contact with human skin for extended periods, and during this time, they can absorb perspiration and sebum. The absorbed secretions can potentially cause changes in the structure of the deteriorating elastomer, ultimately contributing to the overall deterioration of the prosthesis [5,40].

During six months of exposure to aging conditions with artificial sebum, the non-pigmented silicone specimens exhibited a notable increase in ΔE values across all categories of silicone. The control category, which lacked nanoparticles, experienced the most significant shift, reaching a ΔE value of 9.02. These findings suggest that the incorporation of nanoparticles, either alone or as composites, can potentially serve as a protective measure against color degradation in silicone materials when subjected to immersion in sebum.

Furthermore, the presence of fatty acids from sebaceous skin secretions in combination with environmental factors can cause silicone polymers to be partially decomposed by breaking down the bonds within the polymer chains. This degradation process may result in the continuous release of byproducts and alterations in the chromatic properties of the silicone. These observations are supported by the studies conducted by Hatamleh et al. and Al-Harbi et al. [40,45].

The color change (ΔE) in the yellow-pigmented silicone specimens with no nanocomposites (control) was significantly higher when exposed to sweat than that in the other silicone categories. Additionally, the color changes in the 1% chitosan–TiO_2_ and 1% TC specimens were significantly greater than those in the 2% TiO_2_ specimens. On the other hand, the ΔE value of the 2% TiO_2_ specimens was significantly lower than those of the 3% chitosan and control specimens. However, all ΔE values of the yellow-pigmented silicone were within the clinically accepted limit of ΔE < 3.

This finding suggests that immersion in acidic sweat may catalyze crosslinking reactions, leading to the formation of an additional polymer network in the silicone [41]. This contradicts the findings of Han et al. [12], who discovered that yellow silicone pigment mixed with nano-oxides significantly impacted the color stability of an A-2186 silicone elastomer.

The color changes observed in the silicone materials may have been triggered by various factors, as the study by Haug et al. [53] suggested. These factors could include impurities introduced during the manufacturing process, reaction byproducts, initiators, or other mechanisms. Understanding the underlying cause of these transformations could contribute to the development of more stable silicone formulations. By advancing the longevity of prosthetic devices, this research could extend their service life.

Additionally, Kiat-Amnauy et al. [33] highlighted that both the duration of exposure and the specific type of silicone elastomer employed significantly influenced the color stability. This implies that the length of time for which the silicone elastomer was exposed to particular conditions, along with its specific formulation, played a crucial role in determining its color stability.

Considering these factors is critical when creating a maxillofacial prosthesis. The environmental conditions of the Middle East—specifically, in the region of Kurdistan/northern Iraq—are characterized by intense heat, dryness, strong sunlight during the summer, and significant climatic shifts in winter. These conditions present challenges for the longevity and color stability of maxillofacial silicones. Extreme environmental conditions, such as elevated temperatures and ultraviolet radiation, can alter silicone elastomers’ mechanical properties and color stability. Further research is needed to delve into the other impacts of incorporating chitosan–TiO_2_ nanocomposites, such as studies of their antimicrobial capabilities and potential allergenicity through cytotoxicity tests with various types of maxillofacial silicone elastomers over a span of one year. Given the crucial role of mechanical properties in dental and maxillofacial materials, an all-encompassing examination of the investigated maxillofacial elastomers should incorporate both a color stability assessment and rigorous mechanical evaluations. A multidimensional approach of this nature would yield a more holistic view of the material’s comprehensive performance and appropriateness for deployment in a demanding therapeutic context.

## 5. Conclusions

1. The core–shell mixing method was proven to be a suitable methodology for synthesizing chitosan–TiO_2_ nanocomposites, resulting in a novel material with enhanced and unique characteristics.

2. The addition of nanosized particles, whether used alone or as a composite, effectively protected the silicone elastomer from accelerated aging conditions, with the exception of outdoor weathering. The significant color changes observed were primarily induced by UV radiation during outdoor exposure.

3. A remarkable alteration in color was predominantly observed in brilliant-red dry pigments. This substantial color change can primarily be attributed to post-polymerization crosslinking caused by the effects of UV radiation from the irradiating light. Conversely, the yellow and blue pigments exhibited color stability across the various conditions. Nevertheless, the degree of color degradation can vary based on several factors, including the specific silicone elastomer, pigment type, opacifier concentration, and specific aging method employed.

4. In comparison with artificial weathering, outdoor weathering induced more significant changes in the ΔE values across all categories of silicone, regardless of whether they were pigmented or non-pigmented.

5. The findings of this research hold significant potential as a useful guide for different uses within maxillofacial prosthetics and silicone-based medical device sectors. The developers and researchers focused on the creation of RTV silicone products that might benefit from integrating chitosan-TiO_2_ nanocomposites to improve color stability under certain aging conditions, ensuring better aesthetic outcomes for patients over time. However, outdoor weathering remains a challenge for all silicone categories, necessitating further research for enhanced resilience.

## Figures and Tables

**Figure 1 nanomaterials-13-02379-f001:**
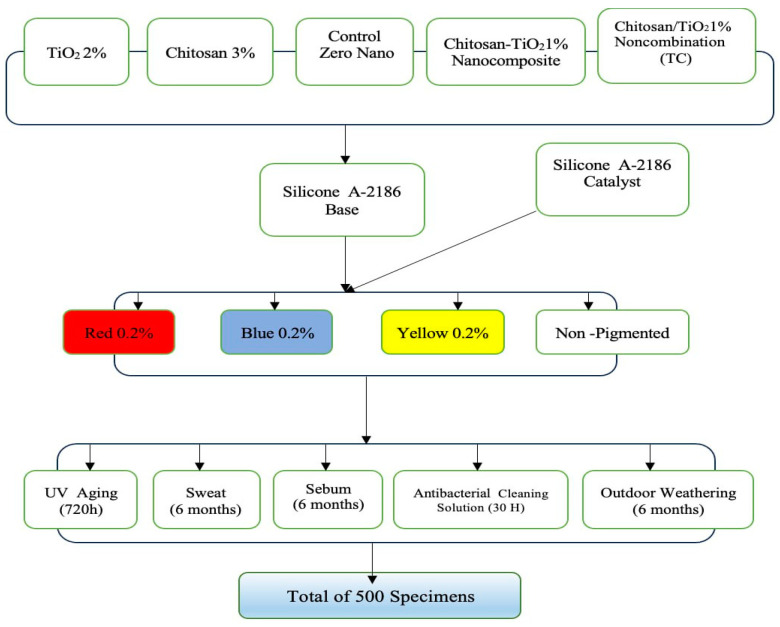
Study design.

**Figure 2 nanomaterials-13-02379-f002:**
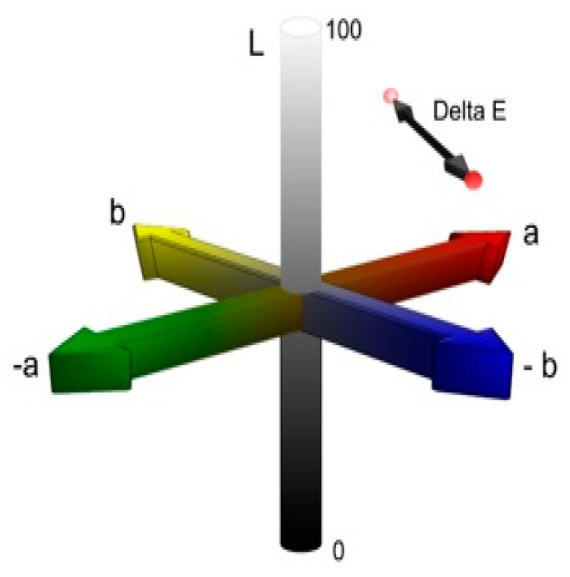
CIELAB color system [5].

**Figure 3 nanomaterials-13-02379-f003:**
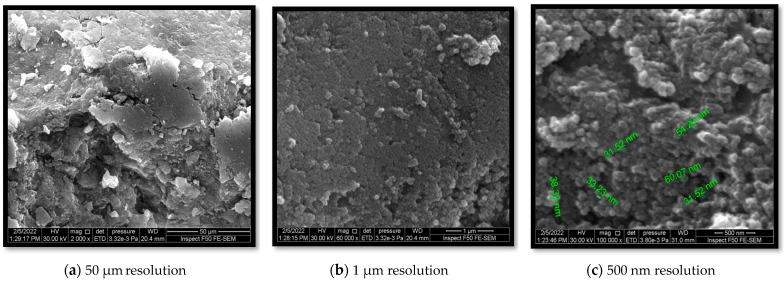
SEM of the synthesized chitosan-TiO_2_ nanocomposite.

**Figure 4 nanomaterials-13-02379-f004:**
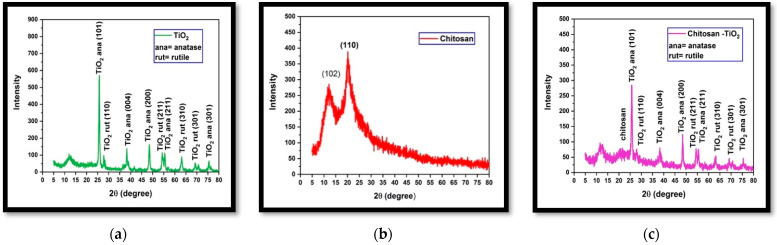
X-ray diffraction: (**a**) XRD of TiO_2_; (**b**) XRD of chitosan (**c**) XRD of chitosan-TiO_2_.

**Figure 5 nanomaterials-13-02379-f005:**
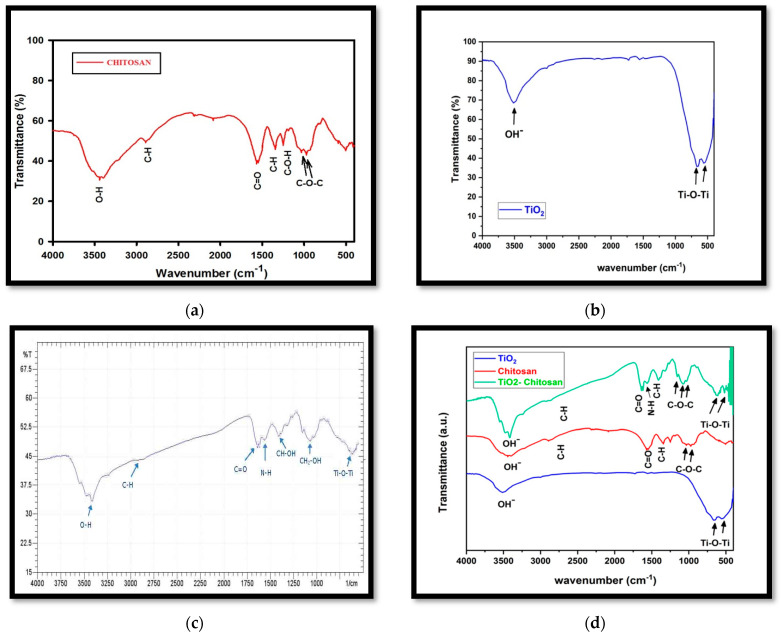
Fourier-transform infrared spectroscopy (FTIR); (**a**) FTIR of chitosan, (**b**) FTIR of TiO_2_, (**c**) FTIR of synthesized chitosan-TiO_2_ nanocomposite, (**d**) FTIR of chitosan, TiO_2_, and chitosan-TiO_2_.

**Figure 6 nanomaterials-13-02379-f006:**
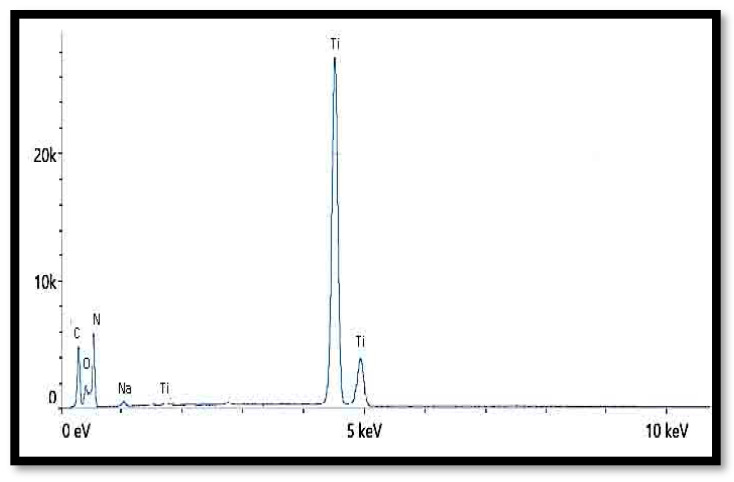
EDX spectra for nanocomposite chitosan-TiO_2_.

**Table 1 nanomaterials-13-02379-t001:** Monthly average climatic data during the outdoor weathering experiment in 2022 in Sulaimani City in the region of Kurdistan in northern Iraq.

Date (2022)	Temperature °C			Average Humidity %	Pressure (mbar)
	Max	Min	Average		
February	15.2	5.4	10.3	64.0	917.4
March	15.3	6.4	10.8	59.6	915.9
April	26.2	14.6	20.4	43.5	914.5
May	28.2	17.3	22.8	41.8	913.2
June	37.4	24.6	31.0	28.3	910.1
July	40.7	26.8	33.7	24.2	906.4
August	42.1	28.0	35.0	23.9	908.5

**Table 2 nanomaterials-13-02379-t002:** EDX results for nanocomposite chitosan-TiO_2_.

Element	Weight %	Atomic %
C	12.9	22.0
N	3.7	5.4
O	42.9	54.8
Na	1.3	1.1
Si	0.4	0.3
Ti	38.8	16.5

**Table 3 nanomaterials-13-02379-t003:** Mean values and standard deviations of ΔE for the brilliant-red-pigmented silicone categories under different conditions (pilot study).

Groups	Brilliant Red ΔE (Pilot Study)			
	1% Chitosan–TiO_2_	1.5% Chitosan–TiO_2_	0.5% Chitosan–TiO_2_	*p*-Value
Sweat (6 months)	6.66 ± 3.60	5.20± 0.36	4.12± 0.87	0.305
Antibacterial cleaning solution (30 h)	3.31 ± 1.67	2.99 ± 2.02	1.28 ± 0.64	0.179
Outdoor weather (6 months)	16.62 ± 6.32 ^b^	17.59 ± 1.68 ^b^	24.64 ± 0.68	0.035
UV weather (1 month; 720 h)	27.92 ± 1.24	26.45 ± 1.11	25.61 ± 2.64	0.162
Sebum (6 months)	6.24 ± 1.66 ^a,b^	10.09 ± 2.96	11.14 ± 0.92	0.01
*p*-value	0.000 *	0.000 *	0.000 *	

^a^: In comparison with 1.5% chitosan–TiO_2_. ^b^: In comparison with 0.5% chitosan–TiO_2_. Different superscript letters in the same row indicate that the significant differences in ΔE (*p* < 0.05) were present only after applying one-way ANOVA, Dennett’s T3 test, and Tukey’s HSD multiple comparison tests. * There were overall significant differences among the conditions in the color changes in all silicone specimens.

## Data Availability

The data that support the finding of this study are available on request from the corresponding author [Al-Kadi, F.K.].

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
