# Peer review of "Hybrid Chitosan–TiO2 Nanocomposite Impregnated in Type A-2186 Maxillofacial Silicone Subjected to Different Accelerated Aging Conditions: An Evaluation of Color Stability"

_nanomaterials, 2023, doi:10.3390/nano13162379_

Round 1
Reviewer 1 Report
The submitted manuscript investigates the effect of Chitosan-TiO2 nanocomposite incorporation on the color stability of pigmented RTV maxillofacial silicone subjected to a variety of accelerated aging conditions. The experimental research is well designed, and the specimens are adequately prepared. The article has the following comments:
1. Introduce the practical application of the core-shell mixing method in the nanocomposite industry.
2. Explain the basic properties of maxillofacial silicone elastomer A-2186.
3. In section 2.2.2 Preparation of Nanocomposite, show a fine powder of the dried composite crushed using mortar.
4. Provide information on aging testing and relevant reference standards.
5. In the conclusion, indicate which applications and options the research results can serve as a reference.
The article needs minor revision for language and grammar.
Author Response
Dear Reviewer, I appreciate your insightful feedback on the manuscript. Your suggestions have been extremely valuable. I have addressed each comment in the detailed response file attached to this communication. Kind Regards

Reviewer 2 Report
A manuscript contain original research work, where the authors discuss result of a study of the influence of inclusion of particles of hybrid TiO2-chitosan (ChT) nanocomposite into the matrix of RVT maxillofacial silicon on color stability of the latter under different conditions of accelerated aging.
I recommend to accept current manuscript, however there are some aspects in this manuscript that should be improved:
1. The morphology of nanocomposites are not sufficiently investigated. It is necessary to perform energy dispercive analysis and evaluate a distribution map of TiO2 nanoparticles in ChT matrix. Size distribution hystograms of TiO2 nanoparticles should be given.
2. When describing the XRD pattern of ChT, authors indicate that reflection peak at 2 theta= 19,77 deg. indicates the amorphous structure of ChT. However, juging from Figure 4b, ChT cannot be called the completely amorphous polymer. Two sharp reflection peaks from plane 110 and 102 indicates the presence of crystalline regions. Therefore, the statement that TiO2 particles are located in amorphous region of ChT requires proof.
3. The IR spectra of nanocomposites should be described in more detail in comparison with spectra of ChT and wether there is an interaction between TiO2 and ChT.
4. There are no data (SEM, XRD, FTIR) in the paper, regarding the distribution of hybrid TiO2-ChT nanoparticles in the structure of silicon polymer.
5. It is not clear why the method for preparing of TiO2-ChT nanocomposite is called "core-shell" method. There is no evidence for the formation nanocomposites with a core-shell structure.
English requires correction errors and typos and careful editing.
Author Response
Dear Reviewer,
I'm grateful for your constructive feedback on our manuscript. Your insights have significantly contributed to its enhancement. I have addressed each of your points meticulously, as detailed in the attached response file.
Sincerely

Round 2
Reviewer 2 Report
The authors have taken unto account almost all the comments of the reviewer. The text has been corrected and supplemented in accordance with reviewer's comments. The reviewer accepted the authors answers and agrees with the included corrections and additions to the text. The article can be accepted for publication.
Minor editing of English language required